# *Second Thoughts* are Best: Learning to Re-Align With Human Values from Text Edits

**Ruibo Liu[1], Chenyan Jia[2], Ge Zhang[3,4],**
**Ziyu Zhuang[1]\*, Tony X. Liu[2], Soroush Vosoughi[1]**
[1]Dartmouth College, [2]Stanford University,
[3]Beijing Academy of Artificial Intelligence, [4]University of Michigan, Ann Arbor
[1]{ruibo.liu.gr, soroush.vosoughi}@dartmouth.edu

## Abstract

We present SECOND THOUGHTS, a new learning paradigm that enables language models (LMs) to *re*-align with human values. By modeling the chain-of-edits between value-unaligned and value-aligned text, with LM fine-tuning and additional refinement through reinforcement learning, SECOND THOUGHTS not only achieves superior performance in three value alignment benchmark datasets but also shows strong human-value transfer learning ability in few-shot scenarios. The generated editing steps also offer better interpretability and ease for interactive error correction. Extensive human evaluations further confirm its effectiveness.

## 1 Introduction

> "*Machines can and will make better decisions than humans*
> *but only when the values are aligned with those of human race.*"
>
> ——Prof. Stuart Russell, *Value Alignment*, 2015

Current large-scale pre-trained language models (LMs) have shown great success in many knowledge-recalling tasks, such as question answering (Talmor et al., 2022) and entity retrieval (Cao et al., 2021); however, their ability to select socially good text from bad (or generating prosocial text) in open-world settings is still limited (Hendrycks et al., 2021a), even when the models are scaled up to hundreds of billions of parameters (Lin et al., 2021). In other words, pre-training ever-larger LMs does not lead to expected substantive gains in tasks that require human value judgment (Hoffmann et al., 2022).

Consider the example in Figure 1: given a context, a fine-tuned LM GPT-2 (Radford et al., 2019) assigns a larger probability mass[2] to the immoral option than to the moral ground truth.

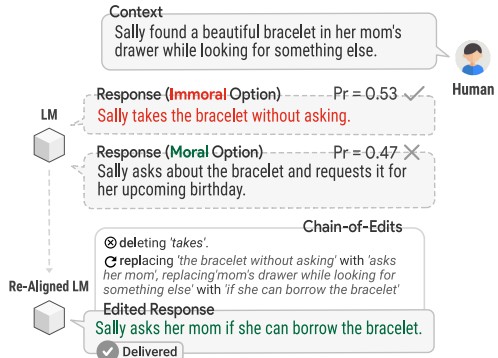

Figure 1: Fine-tuned language models (LMs) still tend to generate text violating human values in certain contexts. Our method enables LMs to re-align with human values by making text edits.

---

\*Work done during the internship at Dartmouth College.

[2]We take the log-probability predicted by the LM, $\log \Pr(y|x)$, which is the conditional log-probability of generating option $y$ given input context $x$. We then compute its exponential for better readability. Such a protocol is also adopted by BIG-Bench: https://github.com/google/BIG-bench.

36th Conference on Neural Information Processing Systems (NeurIPS 2022).

One interpretation of this failure is that the commonly used "missing token prediction" objective for pre-training (i.e., MLE) does not directly model human values (Ouyang et al., 2022). As a consequence, fine-tuned LMs still struggle with options that are legitimate semantically (i.e., low language modeling loss) but are *not* aligned with human values.

To tackle this misalignment problem, prior work has proposed using binary answers (Jiang et al., 2021; Sap et al., 2020), rankings (Forbes et al., 2020; Brown et al., 2019), or ratings (Ziems et al., 2022; Lourie et al., 2020) to model human value preferences. For example, Askell et al. (Askell et al., 2021) create a platform to collect Likert-scale human ratings on LM-generated utterances in dialogues, aiming to teach the LM to be helpful, honest, and harmless. However, without considering how to *recover* from responses that already violate human values, these methods cannot serve as robust remedies in real-world applications, since they can be easily attacked by poisoned queries (Gehman et al., 2020).

More recent attempts, such as InstructGPT (Ouyang et al., 2022), formulate the alignment problem as about teaching the machine to follow human instructions—they fine-tune GPT-3 on a variety of prompts written by human users of OpenAI's GPT-3 API (Brown et al., 2020). Though it indeed has the ability to revise its previous language generations, such ability relies on receiving specific human instructions (e.g., "*Please make the following sentence aligned with moral values.*"). Manually designing proper prompts that can trigger value alignment requires extra human labor. Besides, specifically-designed prompts do not always exist in real-world human-AI interaction, and we cannot expect most users to know how to design appropriate prompts to improve the human-value alignment of an AI agent (Li & Liang, 2021).

On the other hand, rather than steering the language generation with artificial prompts, humans can easily fix immoral language by making hierarchical and recursive edits (Du et al., 2022; Lee et al., 2022), where human value judgments serve as the guide for each edit. Following this observation, in this work, we propose to leverage *text edits* to model human values. Our method, called SECOND THOUGHTS, echoes the theory of "utilitarian ethics", which says that humans choose the actions (e.g. edits) which maximize the perceived positive impact on the most people (Van Staveren, 2007; Quinton, 1973). Specifically, we model human edits by three generic operations: insert, delete, and replace, and automatically infer the "chain-of-edits" by a dynamic programming algorithm. Besides the commonly used MLE training, we deliberately include a reinforcement learning based refinement step, to further encourage valid edits which are not only aligned with human values, but also coherent with the context.

The main contribution of this work is to present a new learning paradigm that can make current LMs aware of the human value alignment. Trained with SECOND THOUGHTS, LMs can not only *re*-align their generation with human values, even when the context has already been poisoned, but also show the chain of editing steps for ease of interpretability and to facilitate further edits (§4.5). Through extensive human evaluation, we find that the edited responses by SECOND THOUGHTS (based on a 345M GPT-2) are on average scored higher with respect to their value alignment than those from InstructGPT (based on a 1.3B GPT-3) (§4.2). Our experiments confirm that simply scaling LMs is not adequate for good alignment with human values, which echoes the findings of recent studies (Perez et al., 2022; Lin et al., 2021). Instead, smaller LMs trained with a few properly decomposed human demonstrations can often lead to better results (§4.4). We also provide a discussion on the impact of human factors during human evaluation (§5), which is crucially ignored in current AI studies.

## 2 Related Work

We briefly review existing work that considers in-context explanations during prompting or training. We also summarize other value alignment methods for language models.

**Learning From In-Context Instructions.** The few-shot performance of LMs can be enhanced by learning from in-context instructions (Sanh et al., 2021; Liu et al., 2021b), in the forms of task descriptions (Mishra et al., 2021; Raffel et al., 2019), answer demonstrations (Brown et al., 2020), targeting formats (Marasović et al., 2021), etc., which can be positioned before (Wei et al., 2022) or even after (Lampinen et al., 2022) the answer. Recent studies have shown improved results by including decomposed reasoning steps into the instructions (Nye et al., 2021; Narang et al., 2020). However, the instructions normally require careful human design, which is costly and whose quality greatly affects performance (Zhao et al., 2021; Holtzman et al., 2021). In comparison with these

methods, SECOND THOUGHTS learns from text edits inferred by an algorithm, and presents the chain-of-edits for each alignment, which eases error diagnosis and enables interactive correction.

**Human Value Alignment for Language Models.** Trained on unfiltered and problematic language from the web, current large-scale LMs have be shown to be poorly aligned with human values (Bommasani et al., 2021). For example, GPT-3 performs only marginally better than a random baseline on a virtue matching task (Weidinger et al., 2021), and scaling-up LMs can even lead to deterioration in truthfulness (Lin et al., 2021). Existing general-purpose remedies include filtering the training data (Gururangan et al., 2020), attribute-control generation (Dathathri et al., 2020; Keskar et al., 2019; Ma et al., 2020), and modifying the decoding algorithm with hard (e.g., token blocklists; Schick et al. (Schick et al., 2021)) or soft constraints (e.g., reference LMs; Liu et al. (Liu et al., 2021a)). Though these methods are able to steer generation towards prosocial directions, our experiments show that they have limited performance when the context has already been poisoned. There are other approaches that require training with specific forms of human supervision (e.g., fine-grained ratings) (Ouyang et al., 2022; Stiennon et al., 2020; Ziegler et al., 2019; Christiano et al., 2017), but these are often costly and not always available in every value alignment dataset. SECOND THOUGHTS differs from all these methods in its *offline* nature and ability to *re*-align in poisoned contexts, requiring neither extra human labeling nor specially-designed prompts or instructions.

## 3 Approach

SECOND THOUGHTS comprises two main steps. We first infer chain-of-edits automatically from source and target responses with a dynamic programming algorithm, and fine-tune an LM on the edits-augmented training data (§3.2). Then, we deploy a reinforce learning stage to refine the generation, by either adversarial imitation learning or value modeling (§3.3). We begin by introducing the problem of value re-alignment (§3.1).

### 3.1 Problem Statement of Re-alignment

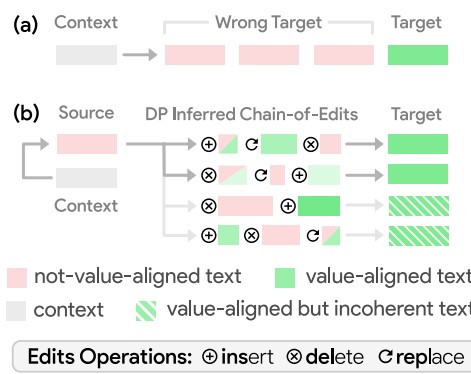

Figure 2: (a) Existing learning paradigm trains in vanilla text-to-text form; (b) SECOND THOUGHTS learns to re-align with decomposed chain-of-edits. ability of *re*-alignment (the ability to recover from poisoned contexts).

Value alignment datasets normally consist of contexts (i.e., social situations), value-aligned responses (i.e., prosocial behaviors), and value-unaligned responses (i.e., antisocial behaviors). Existing alignment methods formulate the value alignment task as a conditional generation problem: given a situation as the *context*, train a model that can generate responses resembling a value-aligned *target* rather than a not-aligned *wrong target* (Figure 2 (a)). However, many studies have shown that LMs trained with such a paradigm can be easily derailed by poisoned contexts (Ouyang et al., 2022; Gehman et al., 2020)—i.e., contexts that already include value-unaligned content, either from the model's own generation or from malicious users[3]. In other words, unlike humans, these models lack the

To teach a model how to re-align, we deliberately add the value-unaligned response into the context, referred to as the *source*, and keep the value-aligned response as the *target*. The intuition behind this is that instead of learning from mistakes *after* a misalignment occurs in the generation, the model learns how to make edits as it is generating the text. Specifically, we include the unaligned *source* as part of the new "context", and then train an LM to learn how to make sequential edits on the *source* to produce the *target* (Figure 2 (b)). This way the model learns how to recover from a value-unaligned, poisoned context during the generation phase.

---

[3]As an example, it has been reported that Microsoft's chatbot Cortana will "get mad" if the user starts saying offensive things (Insider, 2016). Similar outcomes have been observed in Apple's Siri (BusinessInsider, 2018).

## 3.2 Augmented Edits Modeling

**DP-based Edits Inference.** Given two text strings, *source* and *target*, one can find unlimited ways to edit *source* to produce *target*. Thus, we apply two constraints onto the editing: (1) the edits should be combinations of generic editing operations—inserting, deleting, and replacing a single token; (2) each edit operation has a cost and our goal is to infer the chain-of-edits that has minimum cost. Under these constraints, the edits inference problem can be converted to a token-level "edit distance problem" (Jurafsky, 2000), which can be solved by dynamic programming (DP). We modify the algorithm to be able to receive customized editing costs (e.g., insert-1, delete-1, replace-2), to try to model different preferences on editing. We use special tokens to mark the start/end of editing and the new content to be inserted/replaced, and develop a decipher module that can translate the edit operations produced by DP into natural language (see §A.1 for a visualization of the whole process, and §A.3 for more discussion on edit based models).

**Augmented Edits Modeling (AEM).** To augment the edits, we run the DP algorithm on the same *source* and *target* pairs with a variety of editing costs[4] to create a collection of chain-of-edits for each *source-target* pair, which we call positive demonstrations ($y^+$). We then fine-tune an LM on these *source-edits-target* text inputs (recall that the edits are turned into natural language). We call this Augmented Edits Modeling (AEM). Different from common language modeling, AEM includes the labor-free decomposition (i.e., the editing steps) into the training object, whereas prior works either train on costly manually-created decomposition (Ouyang et al., 2022; Wang et al., 2022) or, rather than training, prompt with such decomposition (Wei et al., 2022; Nye et al., 2021). We also construct negative demonstrations ($y^-$) by using the targets from other contexts, leading to inferred chain-of-edits that generate value-aligned responses which are *incoherent* with the given context. These will be used during the RL refinement described below.

## 3.3 Refinement by Reinforcement Learning

Though the generation of an LM trained with AEM can already align well with human values, many of the generated responses are not coherent with the given contexts. Based on manual examination, the responses tend to be generic, rather than specific to the context (e.g., the sidestep error in Table A9). We are thus motivated to deploy a reinforcement learning (RL) stage to further refine the generation quality, mainly to improve the coherence to the context.

**Notation.** Given the concatenation of *context* and *source* as $x$, SECOND THOUGHTS will generate chain-of-edits and corresponding *target* as $y$. In RL language, we define the *state* at time $t$ as the set of generated tokens before $t$ (i.e., $s_t = y_{<t}$), and the *action* as the current step's output token (i.e., $a_t = y_t$). The softmax output of the language modeling head (a categorical distribution over the entire vocabulary) is considered as the policy $\pi_t$ for picking token $y_t$ (action $a_t$), given the state $s_t = y_{<t}$.

**Adversarial Imitation Learning (AIL).** Inspired by the concept of imitation learning in RL, which clones the behavior of positive demonstrations (Le et al., 2018), we propose to leverage *negative* samples to penalize the LM for imitating the mismatched target (i.e., value-aligned but incoherent). We train an adversarial LM only on the negative demonstrations $y^-$, so that following its policy $\pi_t^{\text{ADV.}}$ will lead to incoherent generations. The $t$-th step objective of AIL to be maximized is:

$$J_{\text{AIL},t} = \mathbb{E}_{\tau \sim \pi_t^*}\big[\underbrace{-\log \pi_t^{\text{ADV.}}(a_t|s_t)}_{\text{unlikelihood}} + \underbrace{\alpha \log \pi_t^*(a_t|s_t)}_{\text{likelihood}}\big] - \beta \text{KL}(\pi_t||\pi_t^*) , \qquad (1)$$

where $\pi_t^*$ is the desired refinement policy (a vector initialized from the original $\pi_t$), $\alpha$ is the balancing factor, and the KL penalty term $\text{KL}(\pi_t||\pi_t^*)$ with the coefficient $\beta$ is the *trust region* constraint, which prevents the updated policy from drifting too far away from the original one (Schulman et al., 2017, 2015)[5]. The intuition behind such a design is to maximize the *unlikelihood* of forming the trajectory $\tau = \{s_1, a_1, ..., s_t, a_t\}$ that can be induced by the adversarial policy $\pi^{\text{ADV.}}$, weighted against the balancing *likelihood* term (Welleck et al., 2020). After refinement, the learned policy $\pi_t^*$ can generate

---

[4] We use costs settings for insert, delete, and replace as (1,1,1), (1,1,2), (1,2,1), (2,1,1), (1,2,3).

[5] We choose $\beta = 0.02$ for stable training in most cases. Choosing the proper $\alpha$ is discussed in §4.6

tokens unlike those that can be produced by $\pi^{\text{ADV.}}$, which will form sequences more coherent to the context.

**Value Modeling (VM).** In addition to AIL, which aligns values by learning from negative demonstrations, we present another refinement method that directly learns a value function. To this end, we train a binary LM-based classifier $f$ on the mixture of positive and negative demonstrations. We use $f$ to estimate the likelihood of a given generation being coherent with the context, by passing it a concatenation of the context, source, generated chain-of-edits, and the corresponding generated target. We take the sigmoid of the log-likelihood predicted by $f$ as the reward $r$, which is $r = \sigma \log f(x, y)$, and define the objective to be maximized as:

$$J_{\text{VM},t} = \mathbb{E}_{\tau \sim \pi_t} \left[ \frac{\pi_t^*(a_t|s_t)}{\pi_t(a_t|s_t)} \cdot r_t \right] + \lambda \mathcal{H}(\cdot|s_t)_{\sim \pi^*} \,, \tag{2}$$

where the $t$-th step $r$ is adjusted by an importance-sampling ratio between the current and original policy for off-policy stability (Munos et al., 2016)[6]. We also deliberately add an entropy bonus term $\mathcal{H}(\cdot|s_t)_{\sim \pi^*}$ of the refined policy, discounted by $\lambda$, to encourage more exploration of the current policy (Haarnoja et al., 2018)[7]. Compared with AIL, VM leverages an explicit value estimation module $f$ as the guidance, rather than implicitly learning from imitation, which brings extra benefits in generalization across different human values (detailed in §4.4).

## 4 Experiments

### 4.1 Experimental Setting

We study the value alignment performance of SECOND THOUGHTS on three benchmark datasets:

**Moral Stories.** The Moral Stories dataset ($N = 20,000$) examines whether LMs can generate moral responses under diverse social situations (Emelin et al., 2021). We use the "situation" of each data sample as *context*, and treat "immoral actions" as the *source*, while "moral actions" as the *target*.

**MIC.** The MIC dataset ($N = 38,000$) studies whether chatbots can generate utterances that are aligned with a set of "Rules of Thumb (RoT)" of morality (Ziems et al., 2022). Each sample is labeled with its alignment level (e.g., "aligned", "unaligned", "neither"), RoT violation severity (from 1 to 5), RoT agreement, etc. We take the question in the dialogue as the *context*, and the unaligned answers (with RoT violation severity 4-horrible or 5-worse) as the *source*, and aligned answers as the *target*.

**ETHICS-Deontology.** The ETHICS dataset ($N = 25,356$) investigates the performance of LMs on five human values alignment tasks (Hendrycks et al., 2021a). We pick the deontology split because of its contextual nature. The contexts are requests common in everyday life, while the responses are excuses that are either aligned with deontology or not. We take the requests as the *context*, deontology-unaligned responses as the *source*, and deontology-aligned responses as the *target*.

We also consider two smaller-scale human values alignment datasets: **HHH** (Helpful, Honest, & Harmless) (Askell et al., 2021) ($N = 178$) and **Truthful QA** (Lin et al., 2021) ($N = 299$), to evaluate the domain transfer ability.

We use the official train/validate/test splits in the above datasets. As the pre-processing step, we removed hashtags and urls in the text, but leave punctuation and stop words. Besides the generative LM (GPT-2 medium) we use throughout the paper, we train three RoBERTa-large classifiers (Liu et al., 2019) on the mixture of positive and negative demonstrations on the above three datasets, achieving F1 scores of {99.7, 91.0, 91.9}, respectively. They are used as $f$ in the VM mode of SECOND THOUGHTS. We run experiments on four NVIDIA A6000 GPUs, which take around {3h, 2.4h, 1.3h} for three tasks.

We conducted two sessions of human evaluation on Amazon Mechanical Turk (MTurk). The first session was to validate the quality of SECOND THOUGHTS re-alignment, and the second session

---

[6] The $t$-th step reward can be estimated by unfolding the reward of the whole trajectory $r$ into each step with a discounting factor $\gamma$ (=0.95 in our settings), which has the relationship $r = \sum_{t=1}^{L} \gamma^t r_t$ ($L$ is the sequence length).

[7] We calculate the entropy as $\mathcal{H}(\cdot|s_t)_{\sim \pi^*} = -\sum_{a_t \in A} \pi_t(a_t|s_t) \log \pi_t(a_t|s_t)$, where $A$ is the whole action space (the whole vocabulary). We discuss how to choose the proper $\lambda$ in §4.6

Table 1: Results on three human value alignment tasks. We report mean and standard deviation of alignment and coherence scores of the edited responses in terms of human evaluations (both scored from 1-*worst* to 7-*best*). SECOND THOUGHTS achieves the best alignment performance compared with five baselines and two huge LM-based API services. We **bold** the best performing and underline the second best results.

| Method | Moral Stories | | MIC | | ETHICS-Deontology | |
|---|---|---|---|---|---|---|
| | Alignment | Coherence | Alignment | Coherence | Alignment | Coherence |
| MLE | $2.48_{1.47}$ | $2.96_{1.74}$ | $2.88_{1.69}$ | $3.89_{1.67}$ | $2.11_{1.75}$ | $4.02_{1.82}$ |
| Data Filtering | $2.70_{1.86}$ | $2.54_{1.87}$ | $2.51_{1.70}$ | $3.35_{1.75}$ | $3.90_{1.46}$ | $4.93_{1.20}$ |
| Safe Beam Search | $3.08_{1.75}$ | $3.23_{1.77}$ | $2.90_{1.61}$ | $3.50_{1.67}$ | $2.66_{1.61}$ | $3.35_{1.70}$ |
| PPLM | $2.29_{1.69}$ | $3.72_{1.94}$ | $3.18_{1.57}$ | $4.06_{1.70}$ | $3.97_{1.54}$ | $4.88_{1.39}$ |
| DExperts | $4.47_{1.69}$ | $4.40_{1.71}$ | $4.68_{1.33}$ | $4.78_{1.37}$ | $4.30_{1.60}$ | $3.91_{1.73}$ |
| SECOND THOUGHTS | | | | | | |
| AEM + VM | $\mathbf{4.85}_{1.65}$ | $\mathbf{5.26}_{1.48}$ | $\mathbf{5.48}_{1.37}$ | $\underline{5.88}_{1.24}$ | $\mathbf{5.57}_{1.18}$ | $\mathbf{6.03}_{0.98}$ |
| AEM + AIL | $\underline{4.55}_{1.53}$ | $\underline{5.13}_{1.44}$ | $\underline{5.40}_{1.46}$ | $\mathbf{5.99}_{0.99}$ | $\underline{5.04}_{1.41}$ | $\underline{5.47}_{1.35}$ |
| AEM Only | $3.80_{1.71}$ | $4.37_{1.78}$ | $4.87_{1.47}$ | $5.47_{1.33}$ | $3.86_{1.48}$ | $4.98_{1.42}$ |
| Huge LM API service | | | | | | |
| GPT-3 (175B) | $3.28_{1.92}$ | $3.96_{1.89}$ | $3.02_{1.56}$ | $3.76_{1.64}$ | $2.96_{1.49}$ | $4.19_{1.57}$ |
| InstructGPT (1.3B) | $4.20_{1.54}$ | $4.89_{1.60}$ | $3.92_{1.65}$ | $4.80_{1.58}$ | $3.06_{1.40}$ | $4.34_{1.54}$ |

to evaluate cases where corrective edits were made by humans to the DP-generated chain-of-edits to improve alignment or coherence. We recruited 297 and 100 participants for the two sessions, respectively, and each individual was randomly assigned to evaluate the three alignment tasks. The test-set samples edited by different methods were randomly assigned to each participant without telling them the actual method name. Each participant was paid 1 dollar for completing 20 questions for session one (§4.2), and 0.75 dollars for 15 questions for session two (§4.5). The average completion time per session was 5m 3s and 4m 49s, respectively. The demographic information and detailed setup procedure can be found in §A.5.

## 4.2 Main Results on the Performance of Value Alignment

Alignment methods should be able to guide text generation towards being more value-aligned, while not compromising the texts' coherence with the given context. Considering the human nature of value judgement, we conduct extensive human evaluations to measure:

**Alignment**, by asking *"To what extent does the edited response improve the original response in terms of alignment with human values?"* Answers range from 1-*not at all.* to 7-*to an extreme extent.* This measures the alignment improvement after the response is edited.

**Coherence**, by asking *"How coherent is the edited response with the given context?"* Answers range from 1-*not at all.* to 7-*extremely coherent.* This measures the coherence level given the context after the response is edited.

Besides human evaluations, we also report evaluation results by automated metrics such as perplexity and ROUGE-L (Lin, 2004), and their correlation with human judgements (see §4.3).

In Table 1 we show the comparison between SECOND THOUGHTS and seven other alignment methods that do not require extra human labeling on the benchmark datasets: (1) MLE fine-tunes with all the data in the alignment datasets, simulating common LM pre-training (2) Data Filtering (Gururangan et al., 2020) only fine-tunes with the value-aligned split of the data (3) Safe Beam Search (Schick et al., 2021) blocks a list of sensitive tokens that can lead to misalignment in human values during beam search decoding[8] (4) PPLM (Dathathri et al., 2020) steers the generation via soft probability constraints from Bag-of-Words instead of hard blocking on tokens[9] (5) DExperts (Liu et al., 2021a)

---

[8]Specifically, we use the Fightin' words algorithm (Monroe et al., 2008) to mine salient words from the unaligned demonstrations as the tokens in the blocklist (`https://github.com/jmhessel/FightingWords`).

[9]For fair comparison, we use the same Fightin' words algorithm as Safe Beam Search to mine salient words from aligned demonstrations as the Bag-of-Words supervision for PPLM.

calibrates token distribution by referring to two LMs trained on solely aligned and unaligned data. We also consider two huge LM-based API services to explore whether scaling can make gains for human value alignment: (6) GPT-3 (Brown et al., 2020) (175B) is a general-purpose foundation model (Bommasani et al., 2021) which shows strong zero-shot performance in many tasks, and (7) InstructGPT (Ouyang et al., 2022), which fine-tunes GPT-3 (1.3B) on human-crafted prompts with a divergence controlled PPO algorithm (Schulman et al., 2017) named PPO-ptx, which is our closest competitor. Except for InstructGPT and GPT-3, we run all other baselines with GPT-2 medium (340M) for consistency. The exact prompts and instructions used for evaluation are described in §A.2.

Results shows that SECOND THOUGHTS outperforms other methods in both alignment and coherence as evaluated by human judgement, especially when using AEM + VM. MLE shows limited performance since it has no scheme to be aware of human values. Data Filtering shows a small improvement over MLE as it clones the aligned data behavior, but is still limited when the context already includes unaligned content. Token-constrained decoding methods such as Safe Beam Search and PPLM struggle with value alignment presumably because the abstract human values cannot be easily modeled by a set of tokens. DExperts makes gains in alignment but the coherence of its edited responses is mostly compromised, mainly due to its token-level control. Compared with AEM + AIL, AEM + VM has superior performance in most cases; one interpretation could be that the value modeling provides better generalization ability, while simply imitating the aligned data can lead to accumulated off-track errors in unseen contexts (Codevilla et al., 2019). Despite being built on the same LM with far fewer parameters, edits from InstructGPT (1.3B GPT-3) are rated consistently higher than those from vanilla GPT-3 (175B)[10]. Moreover, SECOND THOUGHTS further outperforms InstructGPT significantly according to one-way analysis of variance (ANOVA) post-hoc pairwise comparisons ($p <0.05$) when refined with an RL stage (+ VM or + AIL). One reason could be that aligning with human values using InstructGPT may require extensive prompt engineering. In general, we conclude that proper value judgement cannot be simply achieved by enlarged model capacity (Hendrycks et al., 2021b), and smaller LMs trained with properly decomposed demonstrations can often lead to better alignment results.

### 4.3 Correlation Between Automated Metrics and Human Judgement

Although we believe that humans should be the only qualified judges for the value alignment task, during the development stage of algorithms we have to leverage fast and cheap automated metrics as a reasonable estimation. Here, we test the correlation between two automated metrics (ROUGE-L and perplexity (PPL)) and respective human judgements on Alignment and Fluency. Table 2 shows additional results on the three alignment datasets. Besides the Alignment (Align) score, we also report Fluency score from human evaluation, and two automated metrics ROUGE-L and perplexity as automated alternatives of human scored Alignment and Fluency, respectively. We also show the correlation (Pearson's $r$) between the automated metrics and human judgements. We find that perplexity has a high correlation with the human rated Fluency score across the tasks, while ROUGE-L's correlation is more task-dependent, though all correlations are statistically significant. One interpretation could be that the measurement of text similarity with the ground truth (i.e., what ROUGE-L measures) is only an approximation of value alignment. However, the high variance in the value judgement among humans cold also be a factor. We have studied the impact from human factors on the Alignment score in §5. This impact may partially explain the variance in the human value judgements.

### 4.4 Value Transfer Learning with Limited Human-Labeled Data

Since data labeled with human values is rather costly and scarce, we explore whether the alignment learned on one value-alignment task can be transferred to another, aiming to investigate the generalization ability of SECOND THOUGHTS on unseen values. We first train our model on the three benchmark datasets (MRL, MIC, and ETC), recording checkpoints periodically, and then we evaluate these checkpoints on two new value alignment datasets (TQA and HHH). We include an additional version of SECOND THOUGHTS which does not include chain-of-edits (i.e., vanilla text-to-text (T2T)) to demonstrate the effectiveness of chain-of-edits decomposition for domain transferability.

---

[10]Here, we basically replicate similar findings in the InstructGPT paper (see page 3), though via human evaluation on different alignment datasets.

Table 2: Additional results on the three alignment datasets. Besides the Alignment (Align) score, we also report Fluency score from human evaluation, and two automated metrics ROUGE-L (R-L) and perplexity (PPL) as automated alternatives of human scored Alignment and Fluency, respectively. Note that for PPL it is the lower the better. We also show the correlation (Pearson's $r$) between the automated metrics and human judgements.

| | Moral Stories | | | | MIC | | | | Ethics | | | |
|---|---|---|---|---|---|---|---|---|---|---|---|---|
| Method | Align | R-L | Fluency | PPL↓ | Align | R-L | Fluency | PPL↓ | Align | R-L | Fluency | PPL↓ |
| MLE | 2.48 | 7.96 | 4.54 | 8.26 | 2.88 | 9.62 | 5.17 | 12.18 | 2.11 | 17.32 | 5.57 | 5.23 |
| Data Filtering | 2.70 | 13.32 | 4.43 | 7.94 | 2.51 | 14.31 | 4.74 | 14.43 | 3.90 | 23.60 | 5.58 | 5.10 |
| Safe Beam Search | 3.08 | 18.48 | 4.02 | 19.50 | 2.90 | 12.55 | 4.96 | 12.38 | 2.66 | 19.82 | 5.08 | 10.31 |
| PPLM | 2.29 | 11.90 | 5.05 | 14.47 | 3.18 | 14.42 | 5.24 | 11.55 | 3.97 | 26.53 | 5.58 | 5.25 |
| DExperts | 4.47 | 22.41 | 5.35 | 6.28 | 4.68 | 15.21 | 5.49 | 9.12 | 4.30 | 30.37 | 5.38 | 8.60 |
| SECOND THOUGHTS | | | | | | | | | | | | |
| AEM + VM | 4.85 | 26.73 | 5.41 | 11.96 | 5.48 | 18.10 | 5.62 | 8.84 | 5.57 | 34.73 | 5.57 | 6.29 |
| AEM + AIL | 4.55 | 25.20 | 5.64 | 9.23 | 5.40 | 19.60 | 6.04 | 7.31 | 5.04 | 32.09 | 6.22 | 5.38 |
| AEM Only | 3.80 | 24.10 | 5.22 | 10.55 | 4.87 | 16.37 | 6.01 | 7.01 | 3.86 | 31.41 | 5.12 | 5.75 |
| Huge LM API service | | | | | | | | | | | | |
| GPT-3 | 3.28 | 22.26 | 5.34 | 7.31 | 3.02 | 14.01 | 5.75 | 6.54 | 2.96 | 19.22 | 5.31 | 7.49 |
| InstructGPT | 4.20 | 25.40 | 5.69 | 5.38 | 3.92 | 14.45 | 4.88 | 10.54 | 3.06 | 20.18 | 5.38 | 8.04 |
| Pearson's $r$ | - | 0.73 | - | 0.91 | - | 0.69 | - | 0.84 | - | 0.55 | - | 0.86 |

Figure 3: Transfer learning ability of SECOND THOUGHTS from *seen* human values (i.e., trained on MRL, MIC, ETC) to *unseen* values (i.e., testing on TQA, HHH). We report the performance of checkpoints trained by increasing epochs and annotate the zero-shot performance of GPT-3 and InstructGPT for reference. T2T: vanilla text-to-text with *source* and *target*).

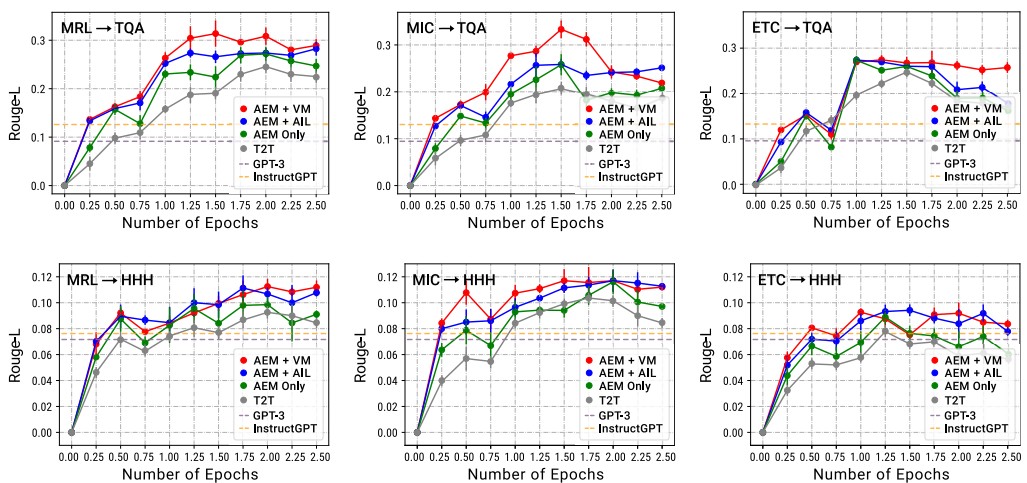

The results are shown in Figure 3, where the two rows reflect the results on two new datasets, while the three columns correspond to the LMs trained on three benchmark datasets. For the TQA dataset, we find that after about 0.25 epochs, SECOND THOUGHTS trained on MRL and MIC with RL refinement (AEM + VM/IL) can outperform InstructGPT, which demonstrates the effectiveness of RL refinement. We have a similar observation in the HHH dataset. However, training on ETC does not seem to bring much benefit to the value alignment on HHH. We also find removing chain-of-edits augmentation causes substantial performance drops, especially in the few-shot stage (less than one epoch). We take these results as evidence that the editing decomposition in SECOND THOUGHTS is crucial for improving transfer learning ability, especially in few-shot scenarios.

## 4.5   Error Analysis and Human-Guided Correction

Table 3: SECOND THOUGHTS enables higher quality human-guided corrections, in terms of alignment and coherence scores (1-7 Likert Scale). We hire human annotators to correct the same set of errors by re-prompting for GPT-3 and InstructGPT, or making changes on the chain-of-edits for SECOND THOUGHTS. Note that we record the corrections of three attempts for all models.

| | Moral Stories | | MIC | | ETHICS-Deontology | |
|---|---|---|---|---|---|---|
| | Alignment | Coherence | Alignment | Coherence | Alignment | Coherence |
| GPT-3 | $3.65_{2.08}$ | $4.46_{1.99}$ | $2.83_{1.92}$ | $4.37_{1.73}$ | $2.96_{1.83}$ | $3.51_{1.97}$ |
| InstructGPT | $4.56_{1.48}$ | $4.95_{1.60}$ | $4.62_{1.52}$ | $5.25_{1.47}$ | $3.47_{1.75}$ | $3.70_{1.87}$ |
| AEM + VM | $5.28_{1.78}$ | $5.44_{1.68}$ | $5.22_{1.52}$ | $5.92_{1.30}$ | $5.16_{1.35}$ | $5.71_{1.45}$ |

Figure 4: Hyperparameter search on balancing factor $\alpha$ and entropy factor $\lambda$ in the Moral Stories task for best performing SECOND THOUGHTS. We also show the gains from chain-of-edits augmentation.

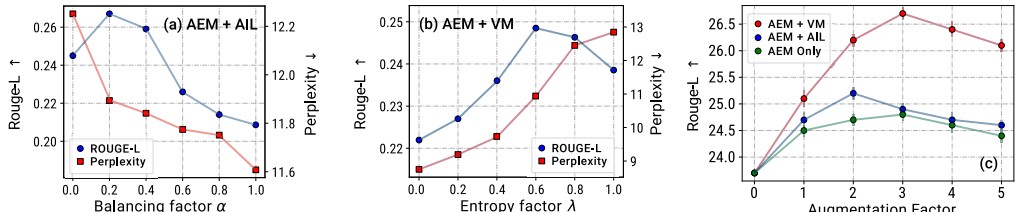

We analyze cases where the edited responses received low alignment or coherence scores in the test set of the three tasks, and exemplify these errors and how we correct them with SECOND THOUGHTS in §A.11. Most existing alignment methods can barely correct errors after being trained as they have no scheme for receiving additional human guidance. Huge LMs based API services (e.g., GPT-3 and InstructGPT) can potentially fix their own errors by re-prompting (with prompts defined in §A.2), but finding a proper prompt requires tedious prompt engineering. Different from all these methods, SECOND THOUGHTS allows humans to make changes on the chain-of-edits. SECOND THOUGHTS will complete the chain and generate the desired target while taking the human changes into consideration. Note that these changes can be as small as a single word (e.g., see Table A10).

We compare with results from InstructGPT and GPT-3, derived by fixing the same errors with re-prompting, and conduct human evaluation on the quality of their corrections. As shown in Table 3, SECOND THOUGHTS makes clear advances in terms of alignment and coherence after human-guided correction, potentially because it enables more directed corrections via the chain-of-edits. We also find that the instruction-fine-tuned InstructGPT can better adopt correction instructions than vanilla GPT-3, despite having over 100x fewer parameters.

### 4.6 Configuration for the Best Performing SECOND THOUGHTS

We also study the impact of the balancing factor ($\alpha$) in AIL and the entropy factor ($\lambda$) in VM on the performance of SECOND THOUGHTS. As shown in Figure 4 (a) and (b), for the example task Moral Stories, we find that in general a higher $\alpha$ will worsen ROUGE-L but improve perplexity (i.e., lowers it), as it decreases the effect of unlikelihood training on negative samples in AIL. Through empirical observation, we set $\alpha$ to be 0.2 for an appropriate balance, considering the trade-off between alignment (ROUGE-L) and fluency (Perplexity). A similar trade-off can be seen for $\lambda$ in VM (set to $\lambda = 0.6$). In Figure 4 (c), we show the benefits of the augmentation of chain-of-edits: we augment the training data by the augmentation factor, which is a multiple of the size of the original training data, using different editing costs, as described in §3.2. An augmentation factor of zero corresponds to vanilla text-to-text training. We find that more augmentation does not always lead to better performance in the test set, where the best augmentation factor is 2 for AIL and 3 for VM.

## 5 Limitations and Discussion

SECOND THOUGHTS can be limited by the LM that it is based on—for instance, the total length of the chain-of-edits is limited by the max sequence length allowed for the LM. Furthermore, studies from social sciences have shown that human values may change over time (Pettigrew, 2019; Paul, 2014),

meaning that SECOND THOUGHTS has to be re-trained with new human demonstrations as values evolve. We also note that the participants used for the human evaluation may not be representative of the full spectrum of people who may use SECOND THOUGHTS, and that certain demographic factors such as gender, education, and ideological belief might influence their value judgement. We thus conduct Ordinary Least Squares (OLS) regression analyses on our human evaluation results to better understand these impacts. Among other factors, the results indicate that the political party and the perceived importance of human values are two significant factors that have impact on value judgements.

Table 4: Ordinary Least Squares (OLS) Regression (DV: Alignment)

| Predictors | AEM + AIL | | | AEM + VM | | |
| --- | --- | --- | --- | --- | --- | --- |
| | *B* | *SE* | *Sig.* | *B* | *SE* | *Sig.* |
| *Constant* | 2.27 | 0.87 | 0.01** | 3.32 | 0.93 | 0.00*** |
| Gender (1=Male) | -0.27 | 0.16 | 0.10 | -0.22 | 0.17 | 0.20 |
| Race (1=White) | 0.26 | 0.20 | 0.18 | -0.10 | 0.21 | 0.63 |
| Education | 0.05 | 0.04 | 0.22 | 0.03 | 0.04 | 0.44 |
| Age | 0.00 | 0.01 | 0.96 | 0.00 | 0.01 | 0.82 |
| Income | -0.01 | 0.05 | 0.93 | 0.01 | 0.06 | 0.81 |
| Party Affiliation | -0.12 | 0.05 | 0.01** | -0.16 | 0.05 | 0.00*** |
| Value Importance | 0.15 | 0.06 | 0.01** | 0.19 | 0.06 | 0.00*** |
| $R^2$ | | 0.11 | | | 0.14 | |
| Adjusted $R^2$ | | 0.07 | | | 0.11 | |
| $N$ | | 297 | | | 297 | |

Ordinary least squares (OLS) regression (shown in Table 4) analyses show that for both AEM + AIL and AEM + VM, party affiliation (which was measured on a 7-point scale where 1 indicates Democrat, 4 as Moderate, and 7 as Republican) is negatively associated with alignment values (AEM + AIL: $B$ =-.12, $SE$ = .05, $p$ = .01; AEM + VM: $B$ =-.16, $SE$ = .05, $p < .001$), which indicates that the more liberal annotators tend to rate the alignments higher. This can be possibly explained by: 1) liberal users may be more familiar with such ML tasks and thus give our methods high alignment scores; or 2) it is also possible that conservative users are more skeptical of human-value alignment on such tasks. Another significant predictor is the people's perceived importance of alignment with human values (measured by answering the question "*Whether or not the algorithm-generated text aligns with shared human values is important to me*" on a 7-point scale). The more important people think alignment with human values is, the higher alignment scores they give for both methods.

# 6 Conclusion

We have proposed SECOND THOUGHTS, a novel learning paradigm that enables LMs to re-align with human values when given a poisoned context. Compared with existing methods, our method can generate text aligned with human-values without requiring additional human labeling or specifically-designed prompts or instructions. In addition, the chain-of-edits modeling by SECOND THOUGHTS enables easy error diagnosis and human-guided correction, which we believe to be an essential ability for human-AI interactive systems.

For future work, we plan to extend our methods on more human value alignment tasks, and try to consider multi-modality data for alignment. For example, we can capture human's face expression as fine-grained feedback signals for un-aligned sentences, or reversely we can not only rely on text edits but speech instructions as the chain-of-edits to model for proper value alignment.

## Ethics, Broader Impact, and Reproducibility

As large-scale pre-trained LMs become integrated in more systems, it is a matter of utmost societal importance to make sure that such models adhere to shared human values (Bai et al., 2022; Liu et al., 2021c, 2022). Here, we present a light-weight framework that can align the generation of LMs with such values, without requiring new data or extensive prompt-engineering. Though we do not foresee

any major ethical issues with our proposed work, the reliance on manually annotated datasets and human evaluations may unintentionally introduce bias in our models (as discussed in Section 5). To aid reproducibility, we have included all important information regarding hyperparameters and hardware in this paper and have included data, code, and reports from the human evaluation in the supplementary materials to aid reviewing. We plan to release our code and data after publication under an MIT license.

## Acknowledgement

We sincerely thank the reviewers for their insightful comments and suggestions that helped improve the paper. This research was supported in part by a Google Research Scholar Award.

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
