# OpenReview forum: "Second Thoughts are Best: Learning to Re-Align With Human Values from Text Edits"
_NeurIPS.cc/2022/Conference — NeurIPS 2022 Accept_

### Official Review · Reviewer_FzTk · 2022-06-27

**Rating:** 5
**Confidence:** 3
**Ethics Flag:** Yes
**Soundness:** 2 fair
**Presentation:** 3 good
**Contribution:** 3 good

**Summary:**

This paper addresses the problem of how we can control large LMs in order to generate text that does not contain any toxic or moral context (i.e., human value-aligned text). In contrast to prior work that focuses on constraining the decoding algorithm or engineering prompts to encourage LMs to produce value-aligned text, the authors propose to perform multiple edits (i.e., insert, delete, replace) iteratively on the generated text until they correct immoral statements. Moreover, instead of generating only one final edited sequence, they generate multiple positive and negative examples and fine-tune the LM given these examples. Finally, the authors include a reinforcement learning refinement step in order to encourage the LM to produce sequences that are value-aligned but still stay in context. They perform human evaluation on text generated by their method and other comparison models and find that the generated text by their approach is more value-aligned and stays in context.

**Questions:**

1. Could you explain in Section 3.2 how exactly you fine-tune the model? What is the input/output, steps and objective?
2. Do you think that your approach could scale to larger LMs? Is there any way to avoid full fine-tuning of the LM? Have you tried more parameter-efficient ways, such as for example adding adapter layers?

**Ethics Review Area:**

["Discrimination / Bias / Fairness Concerns"]

**Limitations:**

As stated in the "Weaknesses" section above, there are two main limitations of this paper. The first is related to the parameters of the LM that have to be fine-tuned, which poses a question about scalability to massive LMs, and the second is related to the human evaluation, which was not performed carefully given the difficulty of this problem.

**Strengths And Weaknesses:**

**Strengths**
1. The authors propose an interesting and intuitive approach for correcting generated text from large LMs. The method is novel and one main advantage of the proposed method is that it is interpretable with multiple edits until they reach the desired text in contrast to prior work that mostly tries to construct prompts or constraint decoding.
2. The authors perform multiple ablation studies and analysis both in the main paper and in the appendix. Overall, their analysis indicates the performance gains of their method, which problems they address in contrast to prior work, and what components of their method are the most important.

**Weaknesses**
1. Although the proposed approach offers text of higher quality in comparison with comparison models, a significant limitation is that the LM has to be fine-tuned multiple times. As far as I understand, the model is first fine-tuned on positive and negative examples generated by editing the source text, and then is further trained via an RL objective. This is an important limitation in comparison to other approaches (constrained decoding, prompts) that do not require re-training of the model. This is especially important when we consider the problem at scale; the authors here fine-tune a 345M model, what will happen when we move to 1.3B or 175B model as the comparison systems? The whole problem at hand is related to how we can control massive LMs that have used the entire web for training and I am not sure whether this method can scale.
2. Section 3.2 is not described in detail, so it is difficult to understand all details. This section would benefit from a more complete description, where the input/output, all steps, and objective for fine-tuning are described.
3. During human evaluation, the authors state that they asked human judges: (a) “to what extend does the edited response improve the original response in terms of alignment with human values?“, and “how coherent is the edited response with the given context”. Although the qualities that the authors try to judge are hard to define in general, the questions are not well formulated. First, for question (a), what are the human values? This is too generic, and it is well-known that not all people share the same values; in fact, there is a large variation. I think this question should be much more specific, stating specific values that the authors want to test (e.g., toxic, racist, or sexist text with examples). Second, question (b) again is too generic, the authors should define coherence, and state specifically and in plain English the qualities that they search for. Given this setup, the results of the human evaluation are not very trustworthy (although there is definitely a positive indication and additional automatic metrics in the appendix that agree with the conclusions of the paper).

---

> ### Author Response · Authors · 2022-08-02
> **Response to Reviewer FzTk**
>
> Thank you for your thoughtful comments and suggestions. We are glad you found our idea interesting and intuitive. We have revised our paper to address your concerns when possible (the general response includes highlights of the revision as well). Below we provide responses to your questions:
>
> **Scalability of Second Thoughts (Weakness 1 / Question 2).**
> Thanks for the great question! We ran additional experiments with our alignment approach on a larger GPT2 checkpoint (GPT2-XL, 1.3B), and Google’s PaLM (Pathway LM) checkpoints 8B and 62B to test the scalability of our method, and the results are shown below (detailed analysis in Section A.9 of our revised Appendix). In general, we see consistent performance gain when using larger LMs. Since Second Thoughts does not change the architecture of the LM (only changing the objective function), we believe there should not be a technical barrier for applying it to larger models.
>
> |                      |  Moral |  Moral |   MIC  |  MIC  | Ethics | Ethics |
> |----------------------|:------:|:------:|:------:|:-----:|:------:|:------:|
> |                      |   R-L  |   PPL  |   R-L  |  PPL  |   R-L  |   PPL  |
> | GPT2-XL (AEM +VM)    | 27.34  | 10.22  | 18.45  | 7.27  | 36.60  |  6.88  |
> | GPT2-XL (AEM +AIL)   |  25.70 |  10.31 |  20.30 |  8.44 |  34.20 |  6.03  |
> | 8B PaLM (AEM +VM)    |  29.70 |  11.59 |  20.57 |  7.31 |  37.17 |  6.40  |
> | 8B PaLM (AEM + AIL)  |  27.34 |  9.33  |  21.04 |  7.29 |  36.56 |  6.56  |
> | 62B PaLM (AEM + VM)  |  31.58 |  9.30  |  23.37 |  6.23 |  38.34 |  5.71  |
> | 62B PaLM (AEM + AIL) |  29.20 |  9.25  |  22.50 |  6.70 |  38.15 |  5.73  |
> | GPT-3                |  22.26 |  7.31  |  14.01 |  6.54 |  19.22 |  7.49  |
> | InstructGPT          |  25.40 |  5.38  |  14.45 | 10.54 |  20.18 |  8.04  |
>
> We believe that performance and parameter-efficiency are trade-offs in the human value alignment tasks. Compared with constrained decoding and prompts based methods, fine-tuning on larger models will undoubtedly be slower. However, as demonstrated in our main results (Tables 1 and A2), Second Thoughts significantly outperforms them, in terms of alignment performance and coherence. Our closest competitor, DExpert [1], fine-tunes two seperate LMs on solely aligned and unaligned data to generate token probability references (detailed in Section 4.2), which we believe requires at least the same level of training cost as Second Thoughts (our RL stage is relatively quick since it does not need to learn the specific domain distribution from scratch but rather refinement). In our preliminary exploration, we have explored prompt-tuning, adapter-based, and froze-LM + meta learning methods, while none of them shows significant and consistent gain on benchmark tasks. Our findings seem to echo the conclusions of a recent study [2], which states that “parameter-efficient methods (e.g., instruction/prompt/adapter tuning) still lag behind full fine-tuning on challenging tasks)”.
>
> In general, we would love to explore other parameter-efficient (but likely lower performing) methods, but it might be out of the scope of this paper (since we take the alignment gain as our first priority, given its societal importance). We agree with the reviewer that it is an interesting direction for future work. We will add a discussion of this to the limitations and future work sections of our camera ready version when we have more space.
>
> **Detailed Training Procedure in Section 3.2 (Question 1 / Weakness 2).**
> Thanks for your suggestions! Given the limited space in the main body, we have added additional details to Section A.1 of the revised Appendix, with clear description of the input, output, training steps, training objective, and training time of the AEM step. Please let us know if any improvements can be made! We will move the more salient parts to the main body in the camera ready.
>
> **Human Annotator Instructions (Weakness 3).**
> Thanks for the careful read of our human evaluation! To show the actual survey we used for human evaluation, we have added a screenshot (Figure A.2) of our MTurk survey in the revised Section A.5. The general questions about Alignment and Coherence are used to inform the participants we are focusing on these two perspectives during the evaluation. However, as we show in Figure A2, we clearly show the participants which specific human value they are going to rate, its definition, and several examples of value-aligned and -unaligned responses. Human participants did not report they had confusions on our evaluation goal.
>
> Finally, we thank the reviewer for the thoughtful comments and constructive suggestions. We hope our response and revision can clarify your confusions and solve your concerns!
>
> [1] [DExperts: Decoding-Time Controlled Text Generation with Experts and Anti-Experts](https://aclanthology.org/2021.acl-long.522.pdf)
>
> [2] [Towards a Unified View of Parameter-Efficient Transfer Learning](https://arxiv.org/pdf/2110.04366.pdf)

---

### Official Review · Reviewer_j2s4 · 2022-07-11

**Rating:** 7
**Confidence:** 3
**Soundness:** 3 good
**Presentation:** 3 good
**Contribution:** 3 good

**Summary:**

This paper proposes a framework that uses sequential editions to align generated text with human values. Specifically, they finetune the language model by creating edit steps between unaligned and aligned sequences using dynamic programming-based methods as augmentation (AEM). The reinforcement learning method is applied to make the generation coherent better with the given context. The experimental results that have better value alignment and coherence compared with many previous language models and language models with larger sizes,

**Questions:**

 Do you think there are more parameter-efficient/post-hoc approaches to tackle this problem (with more tolerance in the performance) as it would be more prevail with service/API-based human value alignment?

Besides, would human interventions (re-labeling, or other adjustments) in the early stage training further help the experiment in Figure 3 (experimental results may not be required for this question)?

**Limitations:**

The authors addresses the limitation of max length capacity and evolving of human values.

**Strengths And Weaknesses:**

Strength
- The human value alignment is an important research question and many controlled generation problems can be framed in this way.
- The editing-based framework offers a lighweight way to plug-in value alignment.

Weakness
- The limited data scenario is helpful to this work is helpful, but the demonstration in epochs makes it not very clear whether the few-shot cases fall into the individual efforts (e.g. 16-shot, 32-shot, 64-shot).

---

> ### Author Response · Authors · 2022-08-02
> **Response to Reviewer j2s4**
>
> Thank you for your reviews! We are glad you find that we are solving an important problem, and the proposed editing-based method is light-weight and promising. We have revised our paper to address your concerns wherever possible (please take a look at our general response for highlights of the revision as well). Below we provide responses to your questions and suggestions:
>
> **Presentation of Few-shot Results.**
> We agree that the actual number of shots might be more informative for the readers. In Section 4.1, we have listed the training data size of each task, so for example, 0.25 epoch of the Moral Stories dataset means 5,000 samples (as there are 20,000 instances in the training set). We currently don’t have enough space to include this in the Figure 3 of our revised paper but we have added Table A11 in the Section A.11 of Appendix to show the calculation results.
>
> **Ideas on More Parameter-Efficient Methods.**
> Thanks for your very interesting question! As huge LMs are nowadays mainly available through API services, then we believe the most promising post-hoc remedy would be prompts-based. One recently proposed method is using prompts for debiasing [1], but whether or not it can perform well on aligning with diverse and complicated human values requires further experiments. There are also many other prompt-based methods focusing on other tasks such as reasoning, for instance using an external verifier (e.g., a calculator) [2] or stepwise verifiers [4] to improve huge LM’s prediction on math problems. For value alignment, one idea could be replacing the calculator for math problems with a “verifier for human values''. These ideas are outside the scope of this paper but can lead to very promising future directions. In the camera ready (given the extra space) we will discuss some of these in our future work. A recent survey paper discussing these post-hoc approaches as “language model cascades” [3] might be of your interests as well.
>
> However, note that a recent study has shown that such parameter-efficient methods (e.g., instruction/prompt/adapter tuning) still lag behind full fine-tuning on challenging tasks [5]. In fact, the general-purpose instruction-aligned InstructGPT still uses fine-tuning to follow instructions (see page 2 of [6], “We focus on fine-tuning approaches …”). In general, as expected (and noted by the reviewer) performance and parameter-efficiency seem to be trade-offs on the value-alignment tasks.
>
> **Human Interventions for Re-Labeling.**
> Thanks for your question. Actually, we have already included this type of method which we call Data Filtering in Table 1. For this method, we only train our LM with the value-aligned data (since the creators of the three benchmark datasets have already provided the labels) and we find the performance on re-alignment is limited, mainly because the LM has no idea how to recover from the value-unaligned generation.
>
> Finally, we thank the reviewer for the thoughtful comments and constructive suggestions!
>
> [1] [Auto-Debias: Debiasing Masked Language Models with Automated Biased Prompts](https://aclanthology.org/2022.acl-long.72.pdf)
>
> [2] [Training Verifiers to Solve Math Word Problems](https://arxiv.org/abs/2110.14168)
>
> [3] [Language Model Cascades](https://arxiv.org/pdf/2207.10342.pdf)
>
> [4] [On the Advance of Making Language Models Better Reasoners](https://arxiv.org/pdf/2206.02336.pdf)
>
> [5] [Towards a Unified View of Parameter-Efficient Transfer Learning](https://arxiv.org/pdf/2110.04366.pdf)

---

### Official Review · Reviewer_uWVF · 2022-07-12

**Rating:** 5
**Confidence:** 3
**Soundness:** 3 good
**Presentation:** 3 good
**Contribution:** 3 good

**Summary:**

The paper targets the alignment of existing Language Models to be aligned with Human values and proposes a new learning method that leverages reinforcement learning for refining the generated text to be more aligned with human values. The proposed learning method models insert, delete, and replace as a chain of edits using reinforcement learning and introduces Augmented Edits Modeling (AEM). The paper also proposes the use of 1) Adversarial Imitation Learning (AIL), which leverages negative samples and an adversarial language model for guiding the target language model to generate text which is more coherent to the context (keeping the value-alignment). And 2) Value Modeling (VM): where a Language Model based classifier is trained to judge the coherence between the generated text and the context.

The paper reports experimental results on three benchmarks, Moral Stories, MIC, and ETHICS-Deontology, and compares the proposed method for Alignment and Coherence of the generated text. The paper further reports the results for transfer learning with limited human-labeled data and performs error analysis using human-guided correction. The results portrayed in the paper show the effectiveness of the proposed AEM architecture.

**Questions:**

Qualitative samples which show less correlation with the Alignment and Coherence metric will be interesting to observe for a reader. Highlighting such examples will make the understanding of these metrics more clear and more reliable.


**Limitations:**

The authors highlight the technical limitation of the proposed method regarding the max sequence length allowed for a language model. Moreover, the authors clearly highlight the biases in the human participants' demographic factors, such as gender, education, and ideological belief, which might have influenced their reported value judgment.


**Strengths And Weaknesses:**

Strengths:
* The paper is well motivated and highlights the problem of value alignment in the existing Language models.

* The method proposed in the paper improves the existing language models by aligning them with human values. The reported results on three benchmarks show the effectiveness of the proposed architecture when compared to other text generation approaches.

* The transfer learning results and error analysis make the results of the proposed architecture more concrete and reliable.


Weaknesses:

* The human evaluation design, as highlighted in the limitation section, might be imposing biases on value judgments due to factors like demographics, education, etc.

---

> ### Author Response · Authors · 2022-08-02
> **Response to Reviewer uWVF**
>
> Thanks for reviewing our paper, and providing your valuable feedback! We are glad that you find that our method is well-motivated and sound with respect to the experimental procedure. We have revised our paper to address your concerns wherever possible (please take a look at our general response for highlights of the revision as well). Below we provide responses to your questions:
>
> **Human Factors.** We are glad that you appreciate that we have clearly discussed limitations of our work. As you note, value alignment in current LMs is an important problem and we have to admit that it is inherently a complicated problem.
>
> Unlike many other works that crucially ignore the human factors (such as demographics) in human evaluation, we conduct further experiments to understand the effects of the human factors (Section A.8 in the Appendix) and discuss it in our limitations (Section 5). We believe a thorough and transparent discussion of the limitations imposed by human factors can help readers understand under what conditions our conclusion would be robust, which we think should be encouraged rather than ignored, not only because NeruIPS encourages that (the checklist, and the reviewing criteria), but also to meet the recent calls from the human-centered AI research community [1,2]. To better aid your reviewing, we have included more details on our human evaluations such as the screenshot of our survey (Figure A2) in the revised Section A.5. Again we thank you for reading our paper carefully and the insightful comment!
>
> **Qualitative Examples.** Thank you for your great suggestion! We have added an additional section (see Section A.4, and Table A2) to show cases where the Alignment and Coherence might be less correlated (we show either value-aligned but incoherent or value-unaligned but coherent, or both). Please let us know if any improvement can be made!
>
> Finally, we thank the reviewer for the thoughtful comments and constructive suggestions. We hope our response can provide further clarity!
>
> [1] [Transparent Human Evaluation for Image Captioning](https://aclanthology.org/2022.naacl-main.254.pdf)
>
> [2] [Annotators with Attitudes: How Annotator Beliefs And Identities Bias Toxic Language Detection](https://aclanthology.org/2022.naacl-main.431/)

---

### Official Review · Reviewer_dpiE · 2022-07-13

**Rating:** 6
**Confidence:** 4
**Ethics Flag:** Yes
**Soundness:** 2 fair
**Presentation:** 2 fair
**Contribution:** 3 good

**Summary:**

This paper proposes a method for using text editing for aligning language models to "human values". They compare with zero-shot GPT3 and Instruct-GPT3, in addition to other methods and generally improve performance on their benchmarks.

---

After response: Thank you for your response! The response has largely addressed my concerns and for this reason, I will increase my score to a 6. For a broader audience, I would strongly recommend adding these definitions of the terms that you added in the rebuttal in revision.

Furthermore, I can appreciate how expensive it can be to train the InstructGPT models, however, given this statement:
```Our experiments confirm that simply scaling LMs is not adequate for good alignment with human values, which echoes the findings of recent studies [50, 34]. Instead, smaller LMs trained with a few properly decomposed human demonstrations can often lead to better results (§4.3).```
I would appreciate a clarification given what was mentioned in the response.

Thank you for your hard work!

---

**Questions:**

### Definitions
What do these terms mean?
- Socially good text
- Moral/immoral text
- Human Values
- Problematic text

For all of these terms, I believe that I have some understanding of what you are getting at, but they are vague/open to interpretation --- could they be better defined?


- In the first figure, what is the likelihood when you compare simply changing "without asking" versus "with asking"? Could this difference in your figure be simply due to the wording?

### Related work
Also, using edit operations is not novel, some works which have explored this are:

[Encode, Tag, Realize: High-Precision Text Editing](https://arxiv.org/pdf/1909.01187)

[Unsupervised Text Style Transfer with Padded Masked Language Models](https://arxiv.org/pdf/2010.01054)

[LEWIS: Levenshtein Editing for Unsupervised Text Style Transfer](https://arxiv.org/pdf/2105.08206)

[Learning to Model Editing Processes](https://arxiv.org/pdf/2205.12374) (I understand this was released at time of submission, but it would be good to cite)

[Levenshtein Transformer](https://arxiv.org/pdf/1905.11006)

Related to this, this work should include a section on edit-based models

### Potentially unfair comparison?
- I don't think it's fair to claim that instruction tuning is inherently worse --- it is possible that if the 1.3B models were finetuned solely on the tasks in question using instructions, then they would perform as well. A more accurate claim, would be that the precision of this method is likely to be better than the large LM models. Furthermore, these prompts use words that are ill-defined/uncommon (morality/deontology) which may contribute to the lack of performance

### Minor comments
:%s/TrustfulQA/TruthfulQA/g




**Ethics Review Area:**

["Discrimination / Bias / Fairness Concerns"]

**Limitations:**

Yes, they provide an ethics statement.

**Strengths And Weaknesses:**

Strengths:
- I genuinely like the method, I think it's simple and has potential for effectiveness
- Mostly well-written

Weaknesses:
- I would appreciate more experiments on other datasets/tasks to verify the applicability of your method.
- Many terms are used (e.g. "human values" without a clear definition to what that refers to in the text)

---

> ### Author Response · Authors · 2022-08-02
> **Response to Reviewer dpiE**
>
> We thank the reviewer for their thoughtful comments and suggestions! We are glad that you like our method and find it simple and effective. We have revised our paper to address your concerns when possible (see general response for highlight of the revision as well). Below we respond to your questions:
>
> **Experiments on Other Value Alignment Datasets.**
> Thank you for the great suggestion! We ran Second Thought on three new datasets (Movie Dic, Cornell IMDB, and DSTC-8 Reddit). The results are shown below (detailed analysis in the newly added Section A.7). We find consistent gains by Second Thought for all datasets.
>
> |                    | Movie Dic | Movie Dic | Cornell IMDB | Cornell IMDB | DSTC-8 | DSTC-8 |
> | ------------------ | :-------: | :-------: | :----------: | :----------: | :----: | :----: |
> | Second Thought     |    R-L    |    PPL    |     R-L      |     PPL      |  R-L   |  PPL   |
> | AEM + VM (default) |   17.35   |   9.23    |    22.47     |     8.84     | 12.56  | 12.40  |
> | AEM + AIL          |   15.02   |   11.96   |    19.60     |     7.31     | 11.31  | 12.85  |
> | AEM Only           |   14.00   |   10.55   |    16.37     |     7.01     |  9.80  | 11.56  |
> | GPT-3              |   10.26   |   10.44   |    11.22     |     8.43     |  7.31  | 11.44  |
> | InstructGPT        |   11.47   |   11.58   |    12.53     |     8.78     |  8.80  | 10.57  |
>
> **Definition of Terms.**
>
> - **Socially good text**: Text that describes socially acceptable behavior, meaning that it aligns with given human values.
> - **Moral/immoral text**: One of our benchmark datasets (Moral Stories) studies the human value “Morality” , and provides a “moral” and an “immoral” option for a given context (shown in Fig 1). In this task, “moral option”, “moral text”, “socially good text”, “value-aligned text” all have the same meaning.
> - **Human Values**: The virtues that guide us when we interact with others. A more formal definition is: certain codes of conduct put forward by a society, and accepted by anyone who meets certain intellectual and volitional conditions [1]. Similar terms include human ethics [1], social norms [2], and societal rule-of-thumbs [3]. There are different types of human values as shown by the different datasets in our paper.
> - **Problematic text**: Text that is not fully aligned with human values. We use “problematic language” in Sec 2 to refer to the unfiltered text corpora that may include value-unaligned and antisocial text, such as hateful speech.
>
> If you think these definitions are useful we will add them to the paper.
>
> **More Related Work.**
> Thank you for the suggested references! Given the limited space, we discuss edit-based models in Sec A.3 .
>
> **Question: Replacing “without” by “with” in Figure 1.**
> Thanks for your question. Fig 1 shows an actual example from the Moral Story dataset, and “moral option” and” immoral option” are two dataset attributes. By changing “without asking” to “with asking”, we are left with two moral options, so no matter what the LM picks, it will be moral. In other words, morality will no longer be a factor. This paper tries to investigate whether an LM can reliably pick value-aligned (moral) text over non-value-aligned (immoral) text and so presenting two value-aligned options is not relevant to our task. However, as was requested, we did run the experiment, changing “without” to “with”: The LM picked the second option with a probability of 0.61.
>
> **Question: Potentially Unfair Comparison.**
> Thanks for your question. We do not claim that instruction tuning is inherently worse; instead, we say it is effective but has limitations (see Sec 1, line 38) since providing proper instruction requires careful engineering and is costly in terms of human labor as new instructions need to be created for each task (e.g., InstructGPT hired 40 people to write prompts). Such carefully-engineered instructions cannot be expected to always exist in real-world human-machine interaction scenarios. We agree that an interesting comparison would involve fine-tuning InstructGPT on the tasks, however, such fine-tuning is not possible for us as the full model has not been made publicly available and can only be accessed via an API.
>
> Our work presents learning from edits so that the LM can align its generation with human values _without_ relying on specific instructions (thus removing this bottleneck). In brief, the contribution of our work is to present an alternative learning paradigm to the traditional language modeling, as an intuitive and effective method to align current LMs with human values.
>
> **Typos.**
> Thank you for pointing these out! They have been fixed.
>
>
> [1] [Aligning AI with Shared Human Values](https://arxiv.org/pdf/2008.02275.pdf)
>
> [2] [Can Machines Learn Morality? The Delphi Experiment](https://arxiv.org/pdf/2110.07574.pdf)
>
> [3] [SOCIAL CHEMISTRY 101: Learning to Reason about Social and Moral Norms](https://aclanthology.org/2020.emnlp-main.48.pdf)

---

> ### Author Response · Authors · 2022-08-07
> **Thank you!**
>
> Thank you for the further response and raising our score! We are happy our revisions and response have addressed your concerns. We will definitely add the corresponding content into our final version. We believe the paper has been made much stronger thanks to your suggestions, and we are grateful for your time!

---

### Review · Ethics_Reviewer_57NE · 2022-08-04

**Recommendation:**

While the authors propose an approach to address the fallacy that human values are static, the authors could similarly discuss potential ways to address or mitigate the limitation they raised about the how political affiliation affects assessment of alignment with human values, such as by reviewing annotations for outliers, collecting annotations from raters of diverse opinion, or reviewing particular examples of inconsistent annotations to discuss where fundamental differences in human opinion are likely to challenge the goal of "achieving" alignment with human values.

**Ethical Issues:**

Yes

**Ethics Review:**

The construction of the proposed method assumes ethics and human values are static and consistent a across contexts, while in fact they may vary substantially depending on the perspective of the human asked to assess such alignment.

---

### Review · Ethics_Reviewer_Udnx · 2022-08-05

**Recommendation:**

I think that my concerns can be addressed with two minor changes.

•	Include more information of the representativeness of the participants (US individuals, 18+ years old, etc) in the main body of the text around the evaluation.  At the very least this ensures readers do not have to find the appendix to assess representativeness
•	Mention that this paper does not discuss the broader issue of who gets to decide value systems and that it only addresses the problem of how to align against an established value system.

I think the paper does a very good job already of addressing many ethical concerns and these are the only two remaining ideas I could think of after a few readings.


**Ethical Issues:**

Yes

**Ethics Review:**

First, I want to say that this is an impressive manuscript and it thoughtfully and directly addresses many ethical concerns clearly and includes a well thought out statement on ethics and broader impact. I’ve read the paper multiple times and I appreciate how much was considered in this paper.  My remaining ethical comments are minor.

My primary concern with the paper is related to the quote at the outset of the paper, specifically:

“Machines can and will make better decisions than humans, but only when the values are aligned with those of human race” – S.Russell, 2015

First, while I’m sure unintended, the quote makes it sound like the values of the human race can be characterized in a simple way throughout the entire human race that we can align our models against.  The views and norms of the world are varied and this quote seems to reduce the problem of value alignment in some sense (to me).  How this connects to the paper is that within the main body of the text in the human evaluations, the demographics of the human evaluators are not presented and the discussion about questions of their representativeness is left until the limitations.  In some sense, leaving a somewhat naïve reader to potentially assume that these results could represent the value system of any group. This is a minor criticism given that the paper does have a limited discussion in the limitations section on representation.

I also am unsure if the paper fully discusses its place with respect to power dynamics in society and around the issues of who decides our value systems (i.e., who collects the data, who labels the corrections, who do they represent and in what cultures were they raised).  The introduction of your method states; “Our method, called SECOND THOUGHTS, echoes the ‘utilitarian ethics’, which says that humans choose the actions (e.g., edits) which maximize the perceived positive impact on the most people…” When this is the case, it brings to mind the Latin phrase “Quis custodiet ipsos custodes?”, or “who will guard the guards themselves?”. That is, how do we know the ascribed value systems themselves hold for all groups of individuals? I think the paper needs to do more to discuss the relationship of how value systems can be decided by groups and how there is a potential for abuse when there are marginalized groups with different values than majority groups.  This relates to the conclusion in the limitations section where factors show a difference along political leanings. I think it might not go far enough though

---

### Author Response · Authors · 2022-08-02
**General Response to All Reviewers**

We thank all the reviewers for their time, valuable comments and constructive feedback. We are glad that the reviewers found strengths in our paper’s novelty, experimental procedure, and contribution to the community. By learning from text edits, Second Thought aims to provide an intuitive and efficient learning paradigm that can align current language models with given human values. We have revised the paper according to the suggestions (highlighted in bronze in the paper). We summarize the highlights from the revision below and address each reviewer’s feedback separately as well.

**The applicability of Second Thought on other alignment tasks (Reviewer dpiE).** Besides the three benchmark tasks and the two value transfer learning tasks, we further run experiments on three alignment tasks (Movie Dic, Cornell IMDB, and DSTC-8 Reddit), which focus on content moderation in dialogues. The experimental results are shown in the newly added Section A.7 of the Appendix. We find that Second Thought can be well adapted to other alignment tasks.

**More qualitative examples when Alignment and Coherence are less correlated (Reviewer j2s4).** In the newly added Section A.4 of the Appendix, we show additional qualitative examples where either the Alignment or Coherence are low (or both). The samples are picked based on human ratings. We would be happy to add additional qualitative examples if the reviewers think that is needed.

**More details on human evaluation (Reviewer j2s4, Reviewer FzTk).** To clarify some confusions in the human evaluation, we have revised Section A.5 of the Appendix by adding the screenshot of our survey (Figure A2) and descriptions of the annotation task. We hope this detailed description makes our human evaluation procedure clearer.

**Ideas on more parameter-efficient methods (Reviewer uWVF, Reviewer FzTk).**
We believe that performance and parameter-efficiency are trade-offs in the human value alignment tasks, echoing the conclusions of a recent study, which states that “parameter-efficient methods (e.g., instruction/prompt/adapter tuning) still lag behind full fine-tuning on challenging tasks [1]”. In our main results (Table 1 Human Evaluation, and Table A2 Automated Evaluation in the Appendix), we have evaluated these methods (safe beam search, prompting, PPLM), and shown that Second Thoughts significantly outperforms them, in terms of alignment performance and coherence to the context. In preliminary explorations, we investigated other parameter-efficient methods such as prompt-tuning, adapter, and froze-LM + meta learning. None of them showed significant and consistent gain on all benchmark tasks.

**The scalability of Second Thought (Reviewer FzTk).** To study the scalability of Second Thought on larger LM checkpoints, we run new experiments on GPT2-XL (1.3B) and PaLM (8B, 62B). We present the results and discuss the training cost in the newly added Section A.9 of the Appendix. We demonstrate Second Thought is capable of scaling up and can bring consistent gains when using larger LMs similar to smaller LMs.

**More details on AEM training (Reviewer FzTk).** We have revised Section A.1 to add more details about our AEM training. We clearly describe the input, output, training steps, training objective, and training time of the AEM step. Please let us know if anything is unclear and we would be happy to add additional detail.

**Minor fixes to the presentation of the paper (thanks to Reviewer dpiE and Reviewer uWVF!).** We have fixed the typos mentioned in the reviews and have added additional clarifying information about our transfer learning experiments in Section A.11 of the Appendix.

We hope our revised paper, additional experiments, and our responses can provide further clarity. Please let us know if there are other things you would like us to address!

[1] [Towards a Unified View of Parameter-Efficient Transfer Learning](https://arxiv.org/pdf/2110.04366.pdf)

---

### Meta-Review · Area_Chair_SPyQ · 2022-08-27

**Recommendation:** Accept
**Confidence:** Less certain

**Metareview:**

This paper received reviews and ratings that are leaning positively, the reviewer discussion and ethics review highlighted weaknesses that make this paper quite borderline.

Strengths:
1. The goal and task of the paper are quite well motivated, and they are geared towards positive societal impact: refining generated text to be more aligned with “human values” (e.g., gearing text towards “moral actions”).
2. The text editing approach of the paper using adversarial imitation learning is both novel and intuitive according to the reviewer.
3. The authors give a convincing justification for their text-edit paradigm, as prior attribute-control generation methods (e.g., PPLM) tend to struggle when the context is polluted. Experimental results appear to support their claim.
4. The chain-of-edit paradigm of the paper (showing, e.g., how the text morphs into a more moral one) eases error diagnosis and enables interactive correction.

Weaknesses:
1. The questions asked to AMT workers in the human evaluation do not seem to be well formulated: (i.e., “To what extent does the edited response improve the original response in terms of alignment with human values?”).  First, the term “human value” is very generic, even the specific human value gets defined later (e.g., “deontology”). The AMT question form reads more like one that would be given to an expert rater, and would need to be either written in plain English or given to trained judges. Second, the judges need to identify *improvement* in alignment with human values could be a source of confusion, as the revised response could align well but no better than the original response (in which case the improvement is non-existent).
2. The ethics reviewers made comments that have bearing on the technical merits of the paper, as they both pointed out that the authors’ modeling assumption that human values are static and consistent across contexts is probably too simplistic. As ethics reviewer Udnx, “views and norms of the world are varied", and may depend on complex contexts (e.g., one may need a lot of background information about a given situation to know which action or statement is more moral). What seems concerning in the paper is that the “context” seems to be always reduced to one sentence, and it seems doubtful this provides enough information to make value judgments in many real-world situations.

In sum, the paper makes valuable contributions, but there may be biases in the results (Weakness 1) and the practical utility may be somewhat limited (Weakness 2). We would recommend that the authors address the AMT-related concern and discuss more extensively their human-value assumptions considering the ethics reviews.

Regarding ethical concerns: we also highly recommend that the authors follow the suggestions proposed by the ethics reviewers, e.g., include more information about the representativeness of the participants.

**Award:**

No

---

### Decision · Program_Chairs · 2022-09-14

Accept